CD14 and CD26 from serum exosomes are associated with type 2 diabetes, exosomal Cystatin C and CD14 are associated with metabolic syndrome and atherogenic index of plasma

Pérez-Macedonio Claudia Paola 1
Flores-Alfaro Eugenia eugeniaflores@uagro.mx 1
Alarcón-Romero Luz del C. 1
Vences-Velázquez Amalia 1
Castro-Alarcón Natividad 1
Martínez-Martínez Eduardo 2
Ramirez Monica mramirezru@conacyt.mx 3
1 Laboratorio de Investigación en Epidemiología Clínica y Molecular, Facultad de Ciencias Químico-Biológicas, Universidad Autónoma de Guerrero , Chilpancingo , Guerrero , México
2 Laboratorio del Metabolismo de RNA y Vesículas Extracelulares, Instituto Nacional de Medicina Genómica (INMEGEN) , México , México
3 CONACYT-Universidad Autónoma de Guerrero , Chilpancingo , Guerrero , México
Sharma Gaurav
Electronic publication date: 2022 Jul 12
Publication date: 2022
Volume: 10
Electronic Location ID: e13656
Received 2021 Oct 18; Accepted 2022 Jun 9
Copyright: ©2022 Pérez-Macedonio et al.
Copyright year: 2022
Copyright holder: Pérez-Macedonio et al.
License: This is an open access article distributed under the terms of the Creative Commons Attribution License, which permits unrestricted use, distribution, reproduction and adaptation in any medium and for any purpose provided that it is properly attributed. For attribution, the original author(s), title, publication source (PeerJ) and either DOI or URL of the article must be cited.
License URL: https://creativecommons.org/licenses/by/4.0/

Keywords: Diabetes, CD14, Cystatin c, CD26, Exosomes, Metabolic syndrome, Atherogenic index of plasma

Funding: Fondo Mixto CONACYT-GUE 249719 INMEGEN 21/2015/I CONACYT-2015 258589 CONACYT-2016 269696 CONACyT-Mexico 705009 This work was supported by the Fondo Mixto CONACYT-GUE (249719); INMEGEN (21/2015/I), CONACYT-2015 (258589), and the infrastructure grant CONACYT-2016 (269696). During the study, CPPM was a grant recipient of CONACyT-Mexico (705009). The funders had no role in study design, data collection and analysis, decision to publish, or preparation of the manuscript.

==============================
Background

Exosomes are microvesicles that actively participate in signaling mechanisms and depending on their content can contribute to the development of different pathologies, such as diabetes and cardiovascular disease.

Objective

The aim of this study was to evaluate the association of cystatin C, CD26, and CD14 proteins in serum exosomes from patients with Type 2 Diabetes (T2D), metabolic syndrome (MetS), and atherogenic index of plasma (AIP).

Methods

Serum exosomes were isolated by ultracentrifugation from 147 individuals with and without diabetes. Both anthropometric and metabolic parameters were registered from everyone. The levels of exosomal proteins cystatin C, CD26, and CD14 were quantified by ELISA. The association between protein levels and T2D or atherogenic risk factors was analyzed by linear regression and generalized regression models.

Results

We observed a significant correlation of increased glucose with elevated levels of Cystatin C, and an effect of T2D on the levels of CD26 (β = 45.8 pg/µg; p = 0.001) and CD14 (β = 168 pg/µg; p < 0.001) compared to subjects without T2D. CD14 was significantly related to T2D, metabolic syndrome, glucose, and the Atherogenic Index of Plasma (AIP). Additionally, we observed a significant effect of metabolic syndrome MetS on the increase of exosomal Cystatin C and CD14.

Conclusions

T2D may contribute to the increase of CD14 protein contained in exosomes, as well as to the predisposition of atherogenic events development due to its relationship with the increase in serum triglyceride concentrations and the AIP score. Finally, the increased levels of CD14 and Cystatin C in exosomes are related to MetS. The analysis of exosome contents of diabetic patients remains an incipient field, so extensive characterization is crucial for their use as biomarkers or to analyze their possible contribution to diabetic complications.

Introduction

Diabetes is a group of metabolic diseases characterized by hyperglycemia caused by defects in insulin secretion or its action. The chronic hyperglycemia of diabetes is associated with long-term damage, dysfunction, and the failure of different organs (American Diabetes Association, 2020). It has been described that the presence of insulin resistance, hypertriglyceridemia, low levels of high-density lipoprotein-cholesterol (HDL-c), increased blood pressure, and central adiposity, are factors that comprise a pathological condition called metabolic syndrome (MetS), a condition that promotes the development of type 2 diabetes (T2D) and cardiovascular disease (CVD) (Akbar et al., 2019). The main components of metabolic syndrome include obesity, high blood pressure, high blood triglycerides, low levels of HDL cholesterol, and insulin resistance (Swarup et al., 2022). According to the International Diabetes Federation (IDF), metabolic syndrome is present if the patient has central obesity plus at least two of the following four factors: ≥triglycerides 150 milligrams per deciliter of blood (mg/dL), reduced high-density lipoprotein cholesterol (HDL-c) less than 40 mg/dL in men or less than 50 mg/dL in women, elevated fasting glucose of ≥l00 mg/dL, or blood pressure values of systolic ≥130 mmHg and/or diastolic ≥85 mmHg (Alberti, Zimmet & Shaw, 2005). Furthermore, T2D is associated with the presence of premature atherosclerosis, which plays an important role in the development of the impaired cardiac function, myocardial infarction, and stroke, therefore, an increase in morbidity and mortality due to CVD (Poznyak et al., 2020; Viigimaa et al., 2020; Kučuk et al., 2021). The atherogenic lipoprotein profile is an important risk factor for coronary artery disease, while the atherogenic index of plasma (AIP) is an index composed of triglycerides and high-density lipoprotein cholesterol. AIP has been used to quantify blood lipid levels and is commonly used as an optimal indicator of dyslipidemia and associated diseases (Zhu et al., 2018).

It has been documented that the presence of biomolecules transported by exosomes has been related to the onset and progression of CVD, thus the emergence of the potential use of exosomes as novel therapeutic targets or biomarkers (Kučuk et al., 2021).

Exosomes are extracellular vesicles (EV) with a size ranging from 30–100 nm, and they are secreted by nearly all cell types under specific pathophysiological conditions. They are the result of the fusion of the multivesicular bodies of endosomal origin with the plasma membrane and have been found in almost all body fluids. It is known that exosomes are key mediators in cell-to-cell communication, acting as transporters of proteins and different classes of RNA (mRNA; lncRNA, and miRNA), or even small amounts of DNA (Kalluri & LeBleu, 2020; Doyle & Wang, 2019; Mathieu et al., 2019; Lu et al., 2019; O’Brien et al., 2020). Their biogenesis mechanism and the selective packaging of proteins, lipids, and several RNA types are not completely understood (O’Brien et al., 2020). Exosomes are characterized by markers such as Alix, HSP70, and the CD9, CD81, and CD63 tetraspanins (Jeppesen et al., 2019). Diverse studies have shown that the number of circulating EVs are increased in patients with insulin resistance, atherosclerosis, obesity, T2D, and vascular complications (Dini et al., 2020; Peng, Liu & Xu, 2020). Some proteins related to an increased risk of cardiovascular disease are transported in microvesicles, including CD26, CD14, and cystatin C, the latter being a protein inhibitor of cysteine proteases (Dini et al., 2020; Peng, Liu & Xu, 2020; Vijay et al., 2018; Yamamoto et al., 2013; Chung et al., 2018). CD26 (dipeptidyl peptidase-4) is a ubiquitously expressed glycoprotein with a catalytic activity that promotes the inactivation of peptides or the generation of new bioactive peptides, thus regulating diverse biological processes (De et al., 2018; Matteucci & Giampietro, 2009; Röhrborn, Wronkowitz & Eckel, 2015). CD26 plays an essential role in glucose metabolism by regulating insulin secretion. This glycoprotein rapidly degrades GLP-1 (a member of the incretin family), thus decreasing glucose-dependent insulin secretion. It is also known that CD26 is elevated in T2D (Drucker & Nauck, 2006). CD14 is a membrane glycoprotein anchored to glycosylphosphatidylinositol (GPI), with a soluble fraction, and it is constitutively expressed in monocytes/macrophages and neutrophils (Liu et al., 2020). Additionally, CD14 levels have been associated with subclinical vascular disease and CVD risk in older adults (Reiner et al., 2013). The relationship among Cystatin C, CD26, and CD14 proteins with atherogenic and cardiovascular risk has already been described in EVs in non-diabetic people and other populations (Kranendonk et al., 2014; Kanhai et al., 2013). In fact, numerous proteins contained in EVs and exosomes that are altered in T2D have been described, but interest was focused on Cystatin C, CD26, and CD14 proteins which have been related to cardiovascular complications in the diabetic population. The studies on the content of exosomes in diabetic people are still incipient therefore is relevant to its analysis. The aim of this study was to analyze the effect of T2D, metabolic syndrome, and the atherogenic index of plasma on the levels of cystatin C, CD26, and CD14 proteins in serum exosomes.

Materials & Methods

Study participants

Seventy-three individuals with and 74 without T2D were enrolled for this study, which was approved by the Ethics Committee of the Autonomous University of Guerrero (#CB-004/17). Participants, men, and women not genetically related, 30–65 years old, were native residents of Guerrero, Mexico, and provided signed informed consent. Measurements of weight, height, waist circumference (WC), and blood pressure (BP) were performed on each participant. Subjects who used lipid-lowering drugs, had excessive alcohol consumption, were active smokers, had any disease different from T2D, or were pregnant women were excluded from the study.

Biochemical assays

Venous blood samples were obtained from participants after a 12 h fasting. Serum levels of glucose, cholesterol, triglycerides, high-density lipoprotein cholesterol (HDL-c), and low-density lipoprotein cholesterol (LDL-c) were determined using enzymatic colorimetric methods with commercially available kits (Spinreact, S.A., Girona, Spain). The atherogenic index of plasma (AIP) was determined using the method proposed by Dobiásová & Frohlich (2001).

Exosome isolation by ultracentrifugation

The blood samples were centrifuged at 2,000× g for 10 min; then the serum was separated and stored at −80 °C until used. Subsequently, 500 µL of serum were, mixed with 700 µL of PBS1x and centrifuged at 120,000× g for 2 h in Optima™ MAX Ultracentrifuge (Beckman Coulter, Life Sciences, IN, USA). The exosome pellet was washed with PBS, centrifuged at 120,000× g for 90 min, and resuspended either with PBS or lysis buffer depending on subsequent application.

Protein quantification assays

The exosome pellet was resuspended in 100 µL of RIPA buffer (25 mM Tris-HCl [pH 7.6], 150 mM NaCl, 1% NP-40, 1% sodium deoxycholate, 0.1% SDS, 1 mM EDTA), and protease inhibitors (Santa Cruz Biotechnology, TX, USA). Protein content was determined using the Pierce™ BCA Protein Assay Kit (Thermo Fisher Scientific, MA, USA). Briefly, standards or the exosome extract were placed in a 96-well plate with 200 µL of BCA working reagent according to the manufacturer’s instructions, and the plate was incubated at 37 °C for 30 min. Subsequently, absorbance at 562 nm was measured on a plate reader DTX-880 (Beckman Coulter, CA, USA).

Transmission electron microscopy (TEM)

A total of 10 µL of exosome samples were loaded on formvar coated carbon grids and incubated for 2 min. The drops were removed, and the exosomes were fixed with 5 µL of 2.5% v/v glutaraldehyde in 0.1M PBS. Then the carbon grid was washed. After removing the water, the grid was stained with 5 µL of 1% uranyl acetate for 1 min. The images were obtained using a JEM-1400 TEM (JEOL, Ltd., Tokyo, Japan). Counting of exosomes was performed using the TEM Exosome Analyzer software as well as the images obtained in TIFF format as described by Kotrbová, et al. (Kranendonk et al., 2014; Kotrbová et al., 2019).

Western blot

A total of 40 µg of total protein of exosome lysate were separated by 10% SDS-PAGE and transferred to PVDF membranes. The membranes were blocked with Tris-Buffered Saline solution containing 0.05% Tween (TBS-T) and 5% non-fat dry milk and incubated overnight with the primary antibodies anti-CD9 (# MA1-80307, 1:5,000; Invitrogen, Thermo Fisher, MA, USA), anti-CD63 (# sc-5275 1:1,000; Santa Cruz Biotechnology, CA, USA) and anti-CD81 (# MA5-13548; Invitrogen, Thermo Fisher, MA, USA) at 4 °C. Then, the membranes were washed three times with TBS-T and incubated with HRP conjugated secondary antibodies (1:30,000 donkey anti-rabbit IgG-HRP sc-2313; 1:50,000 donkey anti-mouse IgG-HRP sc-2314; InvitrogenTM; Thermo Fisher, MA, USA) for 90 min. The membranes were washed again and incubated with an enhanced chemiluminescence substrate (SuperSignal™ West Femto Maximum Sensitivity Substrate; ThermoFisher Scientific, MA, USA). The immunoreactivity signal was revealed by chemiluminescence (ChemiDoc™; Bio-Rad Laboratories, Inc., CA, USA).

Quantification of cystatin C, dipeptidyl peptidase 4 (CD26), and CD14 proteins from exosomes

Protein quantification of exosomes was performed by Human Cystatin C Quantikine ELISA according to the manufacturer’s instructions: (R&D Systems, Minneapolis, MN, USA), Human ELISA Kit DPP4/CD26, and Human ELISA Kit CD14 (Thermo Fisher Scientific, MA, USA) were used to quantify cystatin C, CD26 and CD14, respectively. Briefly, 50 µL of isolated exosomes were resuspended in RIPA, added to each well, and incubated at 4 °C for 3 h; then, the plate was washed and 200 µL of conjugated protein was added and incubated with the substrate. Finally, the stop solution was added to each well, and absorbance was measured at 450 nm in a microplate reader Multiskan™ (GO; Thermo Fisher Scientific, Waltham, MA, USA).

Statistical analysis

For the descriptive statistics, we employed mean values with standard deviations (SD) and medians with interquartile ranges (IQR), for the continuous variable normally and non-normally distributed, respectively. Comparison of frequencies means, and medians between groups was done using Chi-square, Student t, and Mann–Whitney U tests, respectively. The effect of T2D, metabolic syndrome, the number of its components, or of each of the atherogenic risk factors studied on exosomal protein levels was evaluated separately by generalized linear models (GLM), obtaining beta coefficients and 95% confidence intervals, assuming a normal (Gaussian) dispersion. The models were adjusted for the confounding variables of age and gender of the subjects studied. p values <0.05 were considered statistically significant. Statistical analysis was performed using STATA ver. 15.1 statistical software.

Results

Demographic and clinical characteristics of the analyzed population

The average age of the participants was 50 years, individuals with T2D were significantly older, and the percentage of women was higher in the two groups analyzed (86.4%). Of the 73 patients diagnosed with T2DM, 66 had drug treatment and seven had no drugs for disease control. The drugs used by the participants for diabetes control were metformin, insulin, glibenclamide, acarbose, and sitagliptin. T2D patients exhibited a significant increase in systolic blood pressure (SBP) (p < 0.001) and diastolic blood pressure (DBP) (p = 0.049), as well as in serum glucose levels (p < 0.001), cholesterol (p = 0.013), triglycerides (p = 0.036) and low HDL-c levels (p = 0.018). Diabetes patients had a higher AIP (p = 0.005), as well as a significant increase in the levels of the CD14 (<0.001) and CD26 (0.033) proteins (Table 1). A comparison of exosomal protein concentrations between controlled and uncontrolled T2D individuals was performed, however, no significant differences were observed.

Table 1 Demographic and clinical characteristics of study groups.

Data are reported as medians (p25th–p75th) or n (%).

Characteristic	Total n = 147	Without T2D n = 74 (50.3%)	With T2D n = 73 (49.7%)	P	
Gender, n (%)					
Male	20 (13.6)	7 (9.5)	13 (17.8)	0.14a	
Female	127 (86.4)	67 (90.5)	60 (82.2)		
Age (years)	50 (45–56)	48 (42–53)	53 (48–58)	<0.001b	
MetS, n (%)	67 (45.6)	21 (28.4)	46 (63.0)	<0.001a	
AO, n (%)	94 (64)	43 (58.1)	51 (69.9)	0.14a	
SBP (mmHg)	118 (107–129)	114 (104–121)	123 (112–134)	<0.001b	
DBP (mmHg)	71 (64–78)	68 (62–77)	72 (66–80)	0.049b	
Glucose (mg/dL)	94 (82–117)	86 (80–97)	116 (89–205)	<0.001b	
Cholesterol (mg/dL)	218.5 ± 52.2	229.2 ± 48.7	207.8 ± 53.7	0.013a	
Triglycerides (mg/dL)	155 (118–204)	145 (107–187)	170 (129–214)	0.036b	
HDL-c (mg/dL)	47.7 (38.3–56.8)	51 (41.3–59.4)	44.3 (36.5–52)	0.018b	
LDL-c (mg/dL)	132 (103.4–157.4)	135 (103.9–154.7)	129 (101.6–159.1)	0.68b	
AIP	0.16 (0.04–0.32)	0.10 (−0.05–0.25)	0.21 (0.07–0.38)	0.005b	
Low risk (AIP < 0.11), n (%)	60 (42.9)	38 (54.3)	22 (31.4)	0.024	
Intermediate risk (AIP 0.11–0.21), n (%)	20 (14.2)	8 (11.4)	12 (17.1)		
Increased risk (AIP > 0.21), n (%)	60 (42.9)	24 (34.4)	36 (51.5)		
Cystatin C × 10−2 (ng/µg)	98.1 (94.6–102)	98 (94.6–101.6)	102 (94.6–105.1)	0.13b	
CD26 (pg/µg)	78 (44.5–102.5)	67.8 (39.7–101.4)	85.3 (57.5–116.2)	0.033b	
CD14 (pg/µg)	153 (61–286)	97.6 (31–192)	246 (94.3–400)	<0.001b	
Notes.

a Chi-square test

b Mann Whitney

T2D Type 2-diabetes

MetS Metabolic syndrome

AO Abdominal obesity

SBP Systolic blood pressure

DBP Diastolic blood pressure

HDL-c High-density lipoprotein-cholesterol

LDL-c Low-density lipoprotein-cholesterol

AIP Atherogenic index of plasma

Most individuals with T2D had polytherapy consisting of metformin and insulin (36.4%), followed by metformin monotherapy (25.8%) and insulin monotherapy (22.7%), while the remaining percentage consisted of glibenclamide alone and polytherapy.

Exosomes characterization

To confirm the presence of exosomes obtained from the serum of participants with and without T2D, the visualization and identification of exosomes were performed by transmission electron microscopy (TEM). Exosomes showed the characteristic “cup-shaped” morphology, and a variety of sizes was observed (Fig. 1A). Interestingly, exosomes from individuals without T2D had an approximate diameter of 100 nm, while exosomes from individuals with T2D were smaller (50 to 70 nm) (Figs. 1A–1C). However, the former showed a lower number of exosomes per field (2 in average), while the latter showed a higher number (11 exosomes per field in average). (Fig. 1E). We also perform total protein quantification after we verified the presence of the exosomal markers CD63, CD81, and CD9 tetraspanins in both groups (Fig. 1F).

Figure 1 Exosomes characterization.

(A) Exosome characterization. Exosomes were isolated by ultracentrifugation from serum samples and observed by electron microscopy. (A and B) Exosomes from the serum of individuals without diabetes at different magnifications; (C) and (D) Exosomes from the serum of patients with Type 2 Diabetes (T2D) (E) Exosomes quantification. (F) Identification of CD9, CD81, and CD63 proteins in exosomes from serum. Western blot of CD9, CD81, and CD63 constitutive proteins of exosomes. The first lane corresponds to the Molecular Weight Marker (MWM), in the following lanes (1, 2, and 3), there are samples of participants without Type 2 Diabetes (without T2D) are observed, while in the last three lanes, there are the bands of participants with T2D.

Correlation between exosomal proteins and atherosclerotic risk factors

Our population data showed that 28.4% of the individuals without T2D had metabolic syndrome, determined by decreased HDL-c levels (44.3%) and glucose values ≥ 100 mg/dL but less than 126 mg/dL (which is a diagnosis of diabetes), and 34.3% showed and increased risk (AIP > 0.21). Due to this, we decided to perform a correlation analysis in all the individuals studied between exosomal proteins and different atherosclerotic risk factors using Spearman’s correlation. Cystatin C was significantly correlated with CD14 (p = 0.001), with the number of MetS components (p = 0.007), and with serum glucose concentration (p = 0.025). On the other hand, CD14 tetraspanin had a positive correlation with the number of MetS components (p < 0.001), with both systolic and diastolic blood pressure (p ≤ 0.01), serum glucose (p < 0.001) and triglyceride (p = 0.004) levels, and with the atherogenic index of plasma (p = 0.008) (Table 2). However, when performing this same correlation analysis by the study group, significant correlations were only identified in individuals with T2D between Cystatin C levels, CD14 and CD26 and MetS components, and CD14 levels with glucose levels (Table 3), possibly due to a loss in the statistical power of the test. Notwithstanding the significant correlations mentioned before, except for the correlations between the levels of cystatin C and CD14 (r = 0.269), or of CD14 with the number of MetS components (0.364) and with the blood glucose levels (r = 0.376), the other correlations could be considered weak or uncorrelated. However, due to the discussion that exists about the use of cut-off points for correlation coefficient interpretation, some of these cut-off points are arbitrary and inconsistent, therefore it has been suggested to be cautious in their interpretation and this should be done in the context of the investigated problem. It has been shown that in the clinical setting it is not usual to find strong correlations, mainly due to the variability in biological processes  (Schober, Boer & Schwarte, 2018). In this regard, we identified potential clinical relationships between exosomal proteins and cardiovascular risk factors, including the correlation between cystatin C with MetS components, and CD14 with AIP (Fig. S1), relationships that we propose should be further investigated.

Table 2 Correlation between exosomal proteins and different atherosclerotic risk factors in all the individuals studied.

Factor	Cystatin C	CD26	CD14	
	r	p	r	p	r	p	
Cystatin (ng/µg)							
CD26 (pg/µg)	0.144	0.09					
CD14 (pg/µg)	0.269	0.001	0.099	0.25			
Components of MetS	0.233	0.007	0.025	0.78	0.364	<0.001	
BMI, kg/m2	0.065	0.46	0.064	0.47	0.124	0.16	
SBP, mmHg	0.087	0.32	−0.074	0.39	0.220	0.011	
DBP, mmHg	0.055	0.53	−0.056	0.53	0.224	0.010	
Glucose, mg/dL	0.196	0.025	0.023	0.79	0.376	<0.001	
Triglycerides, mg/dL	0.153	0.08	0.007	0.94	0.253	0.004	
Cholesterol, mg/dL	0.148	0.09	−0.046	0.59	−0.055	0.53	
HDL-c, mg/dL	0.010	0.91	−0.097	0.27	−0.111	0.21	
LDL-c, mg/dL	0.063	0.48	−0.004	0.96	0.065	0.46	
AIP score	0.112	0.20	0.053	0.55	0.232	0.008	
Notes.

r Spearman’s regression coefficient

MetS Metabolic syndrome

BMI Body mass index

SBP Systolic blood pressure

DBP Diastolic blood pressure

HDL-c High-density lipoprotein-cholesterol

LDL-c Low-density lipoprotein-cholesterol

AIP Atherogenic index of plasma

Table 3 Correlation between exosomal proteins and different atherosclerotic risk factors by study group.

Factor	Without T2D	With T2D	
	Cystatin C	CD26	CD14	Cystatin C	CD26	CD14	
	r	P	r	P	r	P	r	P	r	P	r	P	
Cystatin (ng/µg)													
CD26 (pg/µg)	0.249	0.05					0.044	0.72					
CD14 (pg/µg)	0.205	0.10	0.041	0.75			0.319	0.009	0.054	0.67			
Components of MetS	0.012	0.92	−0.064	0.61	0.218	0.08	0.393	0.001	−0.060	0.64	0.126	0.31	
BMI, kg/m2	−0.073	0.56	−0.017	0.17	0.137	0.27	0.158	0.20	0.229	0.06	−0.020	0.87	
SBP, mmHg	−0.109	0.38	−0.129	0.30	0.192	0.12	0.200	0.11	−0.199	0.11	−0.023	0.86	
DBP, mmHg	−0.125	0.31	−0.054	0.67	0.186	0.13	0.156	0.21	−0.142	0.26	0.099	0.43	
Glucose, mg/dL	0.172	0.17	0.032	0.80	0.148	0.24	0.084	0.50	-0121	0.33	0.280	0.023	
Triglycerides, mg/dL	0.165	0.19	−0.052	0.68	0.188	0.13	0.106	0.40	−0.019	0.88	0.162	0.19	
Cholesterol, mg/dL	0.125	0.32	0.020	0.87	0.002	0.99	0.197	0.11	−0.085	0.50	0.020	0.87	
HDL-c, mg/dL	0.224	0.07	0.029	0.82	0.007	0.96	−0.181	0.15	−0.189	0.13	−0.098	0.44	
LDL-c, mg/dL	0.167	0.18	0.170	0.17	0.072	0.56	−0.065	0.60	−0.188	0.13	−0.003	0.98	
AIP score	0.016	0.90	−0.060	0.63	0.131	0.30	0.150	0.23	0.108	0.39	0.146	0.24	
Notes.

r Spearman’s correlation coefficient

MetS Metabolic syndrome

BMI Body mass index

SBP Systolic blood pressure

DBP Diastolic blood pressure

HDL-c High-density lipoprotein-cholesterol

LDL-c Low-density lipoprotein-cholesterol

AIP Atherogenic index of plasma

Exosomal protein concentration (CD26, CD14, and cystatin C)

Significant differences were identified between the concentrations of CD26 (p = 0.033) and CD14 (p < 0.001) proteins in individuals with T2D compared with controls (Fig. 2). Otherwise, there were no significant differences in cystatin C concentrations between the two groups analyzed.

Figure 2 Differences in the levels of cystatin C, CD26, and CD14 proteins contained in exosomes by the study group.

A significant increase in CD26 and CD14 levels are shown in individuals with type 2 diabetes (T2D) compared to individuals without T2D.

Relationship between exosomal proteins and atherogenic risk

We analyzed the effect of T2D, metabolic syndrome (MetS), and atherogenic risk on the concentration of exosomal proteins using linear regression models adjusted for age and gender. We identified a significant effect of T2D on the increase in the concentrations of the CD26 (p = 0.001) and CD14 (p < 0.001) proteins, and of the AIP on the levels of the CD14 (p = 0.020) protein.

Additionally, we observed a significant effect of MetS (p = 0.006) and high blood pressure (p = 0.012) on the increase of cystatin C levels, and MetS (p = 0.024) and glucose (p ≤ 0.001) on the increase of CD14 (Table 4).

Table 4 Effect of type 2 diabetes, metabolic syndrome, and atherogenic risk factors on exosomal protein levels.

Factor	Cystatin-C × 10−2 (ng/µg) β (95% CI)	p	CD26 (pg/µg) β (95% CI)	p	CD14 (pg/µg) β (95% CI)	p	
T2D	3.1 (−3.5, 9.7)	0.359	45.8 (18.2, 73.4)	0.001	168 (103, 232)	<0.001	
MetS	8.8 (2.5, 15.1)	0.006	9.3 (−16.8, 35.3)	0.486	77.5 (10.4, 144.6)	0.024	
AO	1.1 (−6.0, 8.2)	0.768	5.4 (−24, 34.9)	0.718	54.2 (−22.4, 131)	0.166	
BP ≥ 130/85 mmHg	9.3 (2.1, 16.6)	0.012	−16.8 (−47.6, 13.9)	0.283	45 (−35.4, 125.4)	0.273	
Glucose ≥ 100 mg/dL	6.0 (−0.4, 12.4)	0.064	5.0 (−21.7, 31.7)	0.714	126 (59.2, 192.7)	<0.001	
TG ≥ 150 mg/dL	2.2 (−3.9, 8.4)	0.477	5.7 (−19.8, 31.1)	0.663	48.5 (−17.6, 114.5)	0.150	
Cholesterol ≥ 200 mg/dL	0.9 (−5.4, 7.2)	0.778	−8.2 (−34.4, 18.1)	0.542	−30 (−98.3, 38.4)	0.391	
HDL-c < 40 M or 50 F mg/dL	3.4 (−3.0, 9.7)	0.297	25 (−1.4, 51.4)	0.063	26.2 (−42.8, 95.2)	0.456	
LDL-c ≥ 130 mg/dL	−3.4 (−9.6, 2.9)	0.294	−2.5 (−29, 24)	0.855	−39 (−107, 29)	0.259	
AIP score	11.8 (−1.2, 24.8)	0.076	46.8 (−8.1, 101.7)	0.095	167.4 (26.5, 308.3)	0.020	
Notes.

Generalized linear models adjusted for age and gender.

β Regression coefficient

CI confidence interval

T2D Type 2-diabetes

MetS Metabolic syndrome

AO Abdominal obesity

BP Blood pressure

TG Triglycerides

HDL-c High-density lipoprotein-cholesterol

M Male

F Female

LDL-c Low-density lipoprotein-cholesterol

AIP Atherogenic index of plasma

Discussion

People with type 2 diabetes have a higher cardiovascular risk compared to non-diabetic and cardiovascular complications attributable to atherosclerosis is a major cause of mortality in diabetic persons  (Viigimaa et al., 2020; Eid et al., 2019; Strain & Paldánius, 2018). Therefore, it is relevant to characterize predictive biomarkers of the development of T2D, as well as cardiovascular complications. Recently, it has been proposed that extracellular vesicles such as exosomes play an important role in the pathogenesis and progression of T2D and CVD. Although the mechanisms by which exosomes can contribute to pathophysiological events are not yet known.

In the present study, we evaluated the effect or association of T2D and atherogenic risk factors with the protein levels of cystatin C, CD26, and CD14 contained in the serum exosomes. Cystatin C is a protein that inhibits cysteine proteases which are constantly produced and excreted by all nucleated cells (Vijay et al., 2018). Some studies show that cystatin C has a close relationship with atherosclerotic disease and that it is also a good marker of glomerular filtration rate, a marker even better than creatinine (Shankar & Teppala, 2011). Our results showed a trend of higher levels of cystatin C in the exosomes of the T2D group compared with the control group. Also, we found a significant increase in this protein with the increase in BP (p = 0.012). This data agrees with the relationship between serum cystatin C and hypertension among adults without clinically recognized chronic kidney disease (Shankar & Teppala, 2011). Metabolic syndrome (MetS) is characterized by several metabolic risk factors and is associated with the development of atherosclerotic cardiovascular disease and T2D in adults. We found a relationship between MetS and exosomal Cystatin C, which agreed with Kranendonk et al. they reported that the high levels of cystatin C contained in EV were associated with a high prevalence of metabolic syndrome (Kranendonk et al., 2014). Several studies indicate that cystatin C is a receptor antagonist of TGF-β, which causes the inhibition of the signaling of this anti-inflammatory cytokine, leading to the inhibition of the protective role of TGF-β. In addition, it is known that an imbalance between the expression of cathepsins and their endogenous inhibitor cystatin C is one of the most important mechanisms in atherogenesis (Prats et al., 2010; Vidak et al., 2019; Arpegård et al., 2008).

The CD26 protein or dipeptidyl peptidase IV in its soluble form (sCD26/DPP-IV) has been shown to play a role in the regulation of glycemia, as well as in the development of atherosclerosis (Cahn, Cernea & Raz, 2016). The increase in CD26 activity in healthy individuals has been proposed as a predictor of metabolic syndrome and insulin resistance, which are necessary factors for the development of T2D. Thus, it is considered a new biomarker for these two conditions (Yang et al., 2014). In this study, individuals with T2D had a significant increase in CD26 (p = 0.001) contained in exosomes compared with the controls. However, we did not find a significant relationship between CD26 protein and other atherogenic risk factors. A possible explanation for this may be that, even though in diabetic individuals an increase in CD26 levels has been found to correlate with insulin resistance, CD26 is a glycoprotein with several functions, among which is the regulation of the activity of chemokines and cytokines, molecules that play an important role in the development of atherosclerosis and cardiovascular disease (Ngetich et al., 2021), and this was not evaluated in this work.

Regarding CD14, we found a significant increase of this protein in exosomes from patients with T2D and MetS, this could be related to the inflammatory state that characterizes both metabolic alterations. We also identified a significant relationship between the AIP score with the CD14 protein. Our results differ from those reported by Kranendonk et al. who found that the levels of CD14 contained in extracellular vesicles were associated with a reduction in T2D development relative risk (Kranendonk et al., 2014). The high levels of CD14 in exosomes from T2D patients may have an impact on the inflammatory activity of adipose tissue and insulin resistance, as has been shown by Fernández-Real et al. (2011), who reported that CD14 is an essential factor for the development of T2D. Also, it is known that CD14 is expressed by monocytes and that these play a crucial role in the inflammation promoted by obesity and insulin resistance (Shitole et al., 2019; De Courten et al., 2016). In addition, CD14 has been linked to the development of atherosclerosis, because the extracellular vesicles secreted by monocytes that express this protein can induce endothelial damage in vitro (Aharon, Tamari & Brenner, 2008).

The presence of CD14 in microvesicles was documented previously by Kanhai et al. (2013) they reported that protein levels of CD14, F2 serpin, and cystatin C in microvesicles were associated with an increased risk for new cardiovascular events and mortality in patients with CVD. Our results suggest that individuals with T2D presented an altered molecular content that may have functional repercussions or indicate altered physiology of cells releasing exosomes to the bloodstream. However, one of the limitations of our study is that we do not know the cell type from which the exosomes present in the bloodstream are derived. Extracellular vesicles are a heterogeneous class of nanovesicles, this heterogeneity is represented by several characteristics such as size, membrane composition, and cargo  (Kalluri & LeBleu, 2020; Doyle & Wang, 2019; Jeppesen et al., 2019). Another limitation is the proportion of women and men, however, this does not represent an issue, since the models were adjusted for age and gender to avoid a biased estimator.

Cystatin C, CD26, and CD14 proteins have been widely described in serum and plasma of individuals with disorders such as diabetes, cardiovascular diseases, and kidney failure, among others  (Yamamoto et al., 2013; Chung et al., 2018; De et al., 2018; Röhrborn, Wronkowitz & Eckel, 2015; Reiner et al., 2013). However, the analysis of these proteins inside extracellular vesicles is scarce. Characterization of the proteomic content of exosomes is relevant and of interest for the study of the functional effect of exosomes on cell communication and the exosome’s role in the development of comorbidities in T2D. The advantages of exosomal proteomic profile analysis are that the biomolecules contained in the exosomes are more stable, and they are preserved to a greater extent since their degradation is prevented. Another advantage is that the exosomal protein content characterization can be performed in more detail, allowing the identification of proteins with potential use as disease biomarkers, which are often in low abundance and readily masked by high abundant proteins in serum or plasma. Likewise, the proteomic profile of exosomes can provide information about the cells of origin. The main disadvantage of using exosomal proteins as biomarkers is that there is not a well-established protocol for their isolation (Raimondo et al., 2011; Mosquera-Heredia et al., 2021; Boukouris & Mathivanan, 2015).

In the present work, we confirmed for the first time the presence of the CD14, CD26, and Cystatin C proteins specifically in exosomes and their relationship to metabolic alterations. On the other hand, the Atherogenic Index of Plasma (AIP) is a good marker of atherogenic dyslipidemia that predicts the risk of CVD because it reflects the relationship between protective and atherogenic lipoproteins (Kranendonk et al., 2014). Our findings revealed a significant difference in the AIP in individuals with diabetes that presented an increased atherogenic risk (>0.210) compared with the controls who presented an intermediate atherogenic risk (0.110–0.209). This is supported by similar data reported by Anjum et al. (2018) showing that individuals with T2D have an increased atherogenic risk (0.73 ± 0.23) and that the increase in serum triglyceride levels and the decrease in serum c-HDL levels comprised the most common lipid abnormalities. Even though our sample size is limited, our findings are robust and reliable, and allow us to confirm the association of CD14 protein with DT2. Furthermore, we found that Cystatin C and CD14 in exosomes are related to MetS. The analysis of exosome contents of diabetic patients remains an incipient field, so the extensive characterization of these types of microvesicles is necessary for their use as biomarkers or to analyze their functional impact in several tissues and therefore contribution to diabetic complications.

Conclusions

In conclusion, our study provides evidence that T2D may contribute to the increase of CD14 protein contained in exosomes, as well as to the predisposition of atherogenic events development due to its relationship with the increase in serum triglyceride concentrations. Also, the levels of CD14 contained in exosomes were associated with systolic and diastolic blood pressure, as well as with serum concentrations of glucose, triglycerides, and the AIP score. Finally, the increased levels of CD14 and Cystatin C in exosomes are related to MetS.

Supplemental Information

Figure S1 Correlation between exosomal protein levels and cardiovascular risk factors

Click here for additional data file.

Supplemental Information 2 Western blot membranes without cut

Uncut and unedited western blot membranes, directly obtained from the Biorad imaging system, the photos include the detection of CD9, CD81 and CD63 proteins in exosomes from the serum of diabetic people and control people

Click here for additional data file.

Supplemental Information 3 Western blot membranes without crop and unedited

Uncut and unedited western blot membranes, directly obtained from the Biorad imaging system, the photos include the detection of CD9, CD81 and CD63 proteins in exosomes from the serum of diabetic people and control people

Click here for additional data file.

Supplemental Information 4 All data used to perform the statistical analysis

Click here for additional data file.

Additional Information and Declarations

Competing Interests

Author Contributions

Human Ethics

Data Availability

The authors declare there are no competing interests.

Claudia Paola Pérez-Macedonio conceived and designed the experiments, performed the experiments, analyzed the data, prepared figures and/or tables, authored or reviewed drafts of the article, and approved the final draft.

Eugenia Flores-Alfaro conceived and designed the experiments, analyzed the data, prepared figures and/or tables, authored or reviewed drafts of the article, and approved the final draft.

Luz del C. Alarcón-Romero conceived and designed the experiments, authored or reviewed drafts of the article, and approved the final draft.

Amalia Vences-Velázquez performed the experiments, authored or reviewed drafts of the article, and approved the final draft.

Natividad Castro-Alarcón performed the experiments, prepared figures and/or tables, authored or reviewed drafts of the article, and approved the final draft.

Eduardo Martínez-Martínez conceived and designed the experiments, performed the experiments, authored or reviewed drafts of the article, and approved the final draft.

Monica Ramirez conceived and designed the experiments, performed the experiments, prepared figures and/or tables, authored or reviewed drafts of the article, and approved the final draft.

The following information was supplied relating to ethical approvals (i.e., approving body and any reference numbers):

This study was approved by Ethics Committee of the Autonomous University of Guerrero #CB004/17.

The following information was supplied regarding data availability:

The raw measurements are available in the Supplementary File.

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
