# Peer review of "CD14 and CD26 from serum exosomes are associated with type 2 diabetes, exosomal Cystatin C and CD14 are associated with metabolic syndrome and atherogenic index of plasma"

_PeerJ, doi:10.7717/peerj.13656_

## Round 0.1 · original submission · Major Revisions

As you see from the referee's comments, the reviewers agreed on the paper's scientific value but also found several methodological, the validity of the findings, and language issues. The paper has not achieved the requisite priority for publication. For further editorial considerations, it is essential to revise the manuscript in response to reviewer comments and
also, submit a point-to-point response to the reviewer comments.

Reviewer 1 ·

Basic reporting

No comment

Experimental design

This manuscript by Pérez-Macedonio et. al. explores serum exosomal proteins viz. cystatin C, CD26 and CD14 from patients with T2DM. The authors have used ELISA to determine protein concentrations and electron microscopy for structural analysis of exosomes. The findings reported in this clinical study are new, however, there are major concerns related to the study design and data that need to be addressed prior to consideration for publication.
Major comments:
1. What was the rationale for selecting and analyzing only three exosomal proteins? The authors need to clearly define the rationale behind their hypothesis. There are several other exosomal proteins that are altered in individuals with T2DM in addition to Cystatin C, CD26 and CD14.
2. What do the authors mean by “MetS Components”? Please include a clear definition of these components and cite literature to substantiate definition.

Validity of the findings

3. A major shortcoming of this study is unequal gender distribution. Since 86.4% of the population is females, the conclusions/findings of this study can only be validated in females and do not apply equally in males. Hence, the authors are encouraged to either increase the number of males to an equal proportion to females or change the title and conclusions specific to female population only.
4. Individuals with T2DM are significantly older and this can be a confounding factor on the data. What are the levels of these proteins (Cystatin C, CD26 and CD14) in serum exosomes of non-diabetic age-matched controls?
5. Figure 1B: It appears that the levels of proteins on western blot are different between individuals with and without T2DM. Therefore, a quantification with an appropriate loading control (or total protein) is required to better understand the data presented.
6. Figure 1A: The authors have concluded that exosomes from individuals with T2DM are smaller. Was there a difference in total number of exosomes in the two groups?
7. Table1: The blood glucose of individuals with T2DM ranges from 89-205 indicating both patients with well controlled and poorly controlled blood glucose. What treatment were these patients on? It would add value to the publication of this group is further classified in 1. Normoglycemic and 2. Hyperglycemic individuals.
8. Table 3: Cystatin-C is not significantly associated with T2DM or AO. Moreover, Cystatin C levels in exosomes are not changed in diabetic individuals. This makes the title misleading and therefore the authors are encouraged to modify title and conclusions accordingly.

Reviewer 2 ·

Basic reporting

The Authors has presented a well written and nicely organized manuscript on levels of CD14, Cystatin C and CD26 in serum exosomes and their association with type 2 diabetes, metabolic syndrome and atherogenic index of plasma. This study is important first step in understanding their use as biomarkers before their extensive characterization. I commend the authors for their extensive data collected for this manuscript. Overall, I enjoyed reading this manuscript; though before publication.

I have the following recommendations that need to be addressed to improve the quality of the manuscript.
In my opinion the authors need to provide more details in the introduction particularly explaining the terms such as atherosclerotic risk factors, atherogenic index of plasma etc., as well as roles of CD14, Cystatin C and CD26 in the pathogenesis of diabetes and CVD.

Experimental design

The authors need to explain the purpose of testing these marker in exosomes rather than testing them in serum or plasma directly. What are the advantages and disadvantages? Is the purpose to show that theses proteins are getting transported throughout the body through exosome if so please explain that in more detail in the manuscript.

The authors mentioned that number of circulating EV are increased in patients with T2D and vascular complications, so I would suggest the authors to include a table or figure showing if there are any differences observed in no. of exosomes isolated in patients with T2D vs without T2D.

Please clarify if the correlation performed in this study was for both subject groups combined or with disease group alone. My suggestion would be to perform two separate correlation for both groups.

Validity of the findings

Even though p-values for some of Correlation between exosomal proteins and different atherosclerotic risk factors is < 0.05 the Spearman's regression coefficient is < 0.25 which is considered to be very weak or no correlation which means minimal relationship between vaiables. Authors need to provide clarification regarding this.

Additional comments

Minor comments:
• In Table 1 the authors did not report cholesterol as medians (p25th-p75th).
• In discussion: fix the spacing issue between line 215 and 216

·

Basic reporting

The paper is well-structured and clearly written. There are a few minor issues that need to be fixed further.
(1)The conclusion section of the abstract contains some of the content of the results section. The conclusion section may need to be more precise and clear.
(2)The relationship of Cystatin C with diabetes/MS/AIP is well defined by previous studies. A short paragraph might be needed added to the introduction to explain why the determination of exosomal cystatin C, CD14 and CD26 was chosen as the detection indicators, and whether there was a correlation between these indicators or some consistent pathophysiological mechanisms.

Experimental design

Overall, this section is well designed. There are two issues that may lead to bias in the analysis. The first is that the two groups of populations are over-represented by women, and the second is that the diabetic group is older. It might be better to discuss it in the discussion.

Validity of the findings

Two small issues may be needed to be further clarified.
(1)In the correlation analysis, CD26 of exosomes was not correlated with blood glucose, lipid profile and metabolic syndrome. However, CD26 is significantly elevated in diabetic patients, and the reasons for this may need to be analyzed in the discussion.
(2)AIP score was calculated from TG and HDL-C. There are two points in Table 3 that may require further clarification. The first is why neither TG nor HDL-C was correlated with CD14, but AIP score was a correlated factor for CD14. The second is that AIP score may not be suitable for inclusion in regression analysis at the same time as TG and HDL-C.

Additional comments

No further comments.

---

## Round 0.2 · Minor Revisions

As you see from the reviewer's comments, the reviewer agreed that the manuscript has improved but also suggested a few minor revisions to further hone the manuscript. It is advised to revise the manuscript and send a point-by-point response to the reviewer for additional editorial consideration.

Reviewer 1 ·

Basic reporting

Please modify Table 1 as per comment below:
7. Table1: The blood glucose of individuals with T2DM ranges from 89-205 indicating both patients with well controlled and poorly controlled blood glucose. What treatment were these patients on? It would add value to the publication of this group is further classified in 1. Normoglycemic and 2. Hyperglycemic individuals.
When comparing controlled and uncontrolled T2D individuals, no significant differences were observed among the exosomal proteins studied, only an increase in CD14 was identified in uncontrolled diabetics, however, it was not significant. Therefore, we conclude that the effect on dysregulation of exosomal proteins is mainly due to diabetes, regardless of diabetic control. On the other hand, when we evaluated a generalized regression model, the effect of uncontrolled diabetes on CD14 increase was not significant. Thus, we considered that it is not necessary to change Table 1, since there was no statistical difference between controlled and uncontrolled diabetic individuals and exosomal proteins, however, we leave it to the reviewer to consider whether Table 1 in the manuscript should be modified.

Experimental design

No further comments

Validity of the findings

No further comments

Reviewer 2 ·

Basic reporting

No Comment

Experimental design

No Comment

Validity of the findings

No Comment

Additional comments

No Comment

---

## Round 0.3 · Minor Revisions

The current manuscript has significantly improved and can be given editorial consideration for publication.

On a minor comment: do you really mean the significant figures as reported in the table from latest rebuttal? Consider referring to the rules for Significant Figures reporting.

---

## Round 0.4 · accepted · Accept

I have no additional comments. The manuscript can be considered for publication.